# Associations of Cooking Salt Intake During Pregnancy with Low Birth Weight and Small for Gestational Age Newborns: A Large Cohort Study

**DOI:** 10.3390/nu17040642

**Published:** 2025-02-11

**Authors:** Tongtong Li, Zhengyuan Wang, Zilin Xiao, Chengwu Feng, Zhuo Sun, Dou Mao, Puchen Zhou, Caimei Yuan, Danyang Zhao, Wanning Shang, Yunman Liu, Changzheng Yuan, Li Hong, Jiajie Zang, Geng Zong

**Affiliations:** 1CAS Key Laboratory of Nutrition, Metabolism and Food Safety, Shanghai Institute of Nutrition and Health, University of Chinese Academy of Sciences, Chinese Academy of Sciences, Shanghai 200031, China; 2Division of Health Risk Factors Monitoring and Control, Shanghai Municipal Center for Disease Control and Prevention, Shanghai 200336, China; 3Department of Clinical Nutrition, Shanghai Children’s Medical Center Affiliated to School of Medicine, Shanghai Jiao Tong University, Shanghai 200127, China; 4School of Public Health, The Second Affiliated Hospital, Zhejiang University School of Medicine, Hangzhou 310058, China; 5Institute of Nutrition, Fudan University, No.130 Dong-An Road, Shanghai 200032, China

**Keywords:** salt intake, pregnancy, low birth weight, small for gestational age, cohort

## Abstract

**Background:** Excessive salt intake has been strongly associated with multiple health conditions, while evidence linking salt consumption during pregnancy and birth outcomes remains limited. We aimed to investigate the association between salt intake during pregnancy and adverse outcomes of birth weight. **Methods:** Our study was based on a prospective cohort study that has followed 4267 mother–child pairs since 2017 in Shanghai, China. Salt consumption was estimated based on the cooking salt and soy sauce from household condiments consumed, weighing measurements over a week, and then categorized into <5.0 (reference), 5.0–10.0, and ≥10.0 g/day. Salt density was calculated as the amount of salt divided by the total energy intake from food frequency questionnaires. Outcomes related to birth weight were defined according to standard clinical cutoffs, including low birth weight (LBW), macrosomia, small for gestational age (SGA), and large for gestational age (LGA). **Results:** Multivariable-adjusted odds ratios (ORs) of LBW were 1.72 (95% CI 1.01–2.91) for 5.0–10.0 g/day salt intake, and 2.06 (95% CI 1.02–4.13) for ≥10.0 g/day, compared to those of <5.0 g/day (*p*-trend = 0.04). For SGA, ORs were 1.46 (95% CI 1.09–1.97) for 5.0–10.0 g/day and 1.69 (95% CI 1.16–2.47; *p*-trend = 0.006) for ≥10.0 g/day. Similarly, the OR comparing the extreme tertile (high vs. low) of salt density was 1.91 (95% CI 1.08–3.36; *p*-trend = 0.01) for LBW and 1.63 (95% CI 1.18–2.25; *p*-trend < 0.001) for SGA. No significant associations were observed for salt intake in relation to macrosomia or LGA. These findings remain stable in all sensitivity and subgroup analyses. **Conclusions:** In this study, habitual cooking salt intake above 5 g/day was associated with increased risks of LBW and SGA, which warrants confirmation by interventional studies.

## 1. Introduction

The World Health Organization (WHO) recommends a salt intake of <5.0 g/day (2000 mg/day of sodium), yet the average daily salt intake globally is estimated to be 10.8 g of salt (4310 mg/day of sodium) [1,2,3]. According to a recent report, the average salt consumption for most countries worldwide exceeds the recommended level [2]. A high amount of sodium consumption is a leading risk factor for hypertension [4,5,6], cardiovascular disease [7,8,9], and mortality among adults [10,11], yet findings on birth outcomes are lacking [12]. Several observational studies found no significant association of sodium in food frequency questionnaires (FFQ) or 24 h urinary samples with birth weight or infants being small for gestational age (SGA) [4,13,14,15], while others suggested that reducing sodium or following diets low in salt during pregnancy was associated with a higher birth weight [16,17,18].

Most existing observational studies estimated sodium intake using FFQ, 24 h recall, spot urines, or single 24 h urine samples. However, self-reported assessments of these methods are subject to recall bias and low accuracy in measuring the salt content of foods when it is added during cooking or at the table, while urinary measures show a large day-to-day variation [19]. China is among the countries with the highest salt intake [3], and the average daily consumption is 11.1 g/day [20]. It has been shown that the majority (82.2%) of dietary salt comes from salt and soy sauce added during home cooking among Chinese people [21], while dietary salt mainly comes from processed foods and ready meals in Western countries (90%), and salt added during home cooking or at the table accounts for about 10% of the daily intake [21,22]. In this regard, measuring salt and other condiments’ consumption in the kitchen may provide further insight into the health impacts of high amounts of sodium exposure among the Chinese population.

Here, we aimed to investigate the associations of cooking salt intake during pregnancy, estimated by weighing household condiment consumption over a week, with infant low birth weight (LBW), macrosomia, SGA, and infants being large for gestational age (LGA) in a cohort of pregnant women in China.

## 2. Methods

### 2.1. Study Population and Design

Starting in 2017, the Iodine Status in Pregnancy and Offspring Health Cohort (ISPOHC) study recruited 5042 pregnant women in Shanghai, China [23]. The study utilized a multistage, stratified random sampling approach to acquire a representative sample. At baseline, participants at 6 to 38 gestational weeks were recruited, with approximately one-third in each trimester. Specifically, each district (16 districts in Shanghai) was subdivided into five distinct sampling areas: east, west, south, north, and central. From each of these sampling areas, a street was randomly chosen, and 40 to 70 pregnant women were selected from each street. At baseline, they completed initial questionnaires and biological sample collection. Dietary data included a validated 69-item FFQ and household condiment consumption measurement over a week. In this study, women were excluded if they subsequently had not delivered a live, singleton baby (*n* = 76), reported an implausibly high or low energy intake (less than 500 kcal/day or more than 5000 kcal/day; *n* = 295), had a missing value on their salt intake (*n* = 232), or had missing values on offspring birth weight (*n* = 172), resulting in 4267 mother–child pairs for analysis.

The ISPOHC study obtained ethical approval from the Ethics Committee of the Shanghai Centre for Disease Control and Prevention. All participants provided informed consent by signing a written document.

### 2.2. Salt Intake and Other Dietary Data

At baseline, cooking salt and soy sauce in the home inventory were weighed and recorded at the start and end of a seven-day sampling period. Daily personal intake of salt was estimated based on the number of meals consumed at home over the same period, and then divided by family size and seven days [24]. The amount of salt in soy sauce was estimated by multiplying the weight of soy sauce with 0.15 [25]. Total cooking salt intake (hereafter referred to as salt intake) was calculated by summing the salt added in its solid form during cooking and the salt from soy sauce. We grouped daily salt intake into 3 categories: <5.0 (reference), 5.0–10.0, and ≥10.0 g/day according to the WHO’s maximum intake recommendation of 5 g/day salt for adults [2]. The secondary exposure was dietary salt density, which is the ratio of daily salt (mg) to total energy intake (kcal). Daily food intake was estimated based on FFQ, and energy (kcal/day) and nutrient intakes were converted using the China Food Composition Database [25].

### 2.3. Outcome Ascertainment

Birth weight in grams and gestational age in weeks were obtained through city-wide medical records. The newborns were classified as LBW (birth weight < 2500 g), normal birth weight (birth weight 2500–4000 g), and macrosomia (birth weight > 4000 g). Additionally, SGA was defined as a birth weight below the 10th percentile of the sex-and-gestational-age-specific birth weight, while LGA was defined as above the 90th percentile, per the neonatal growth standards in China [26]. The analysis for SGA and LGA was limited to infants with a gestational age of 24–42 completed weeks.

### 2.4. Covariate Assessment

Trained staff administered questionnaires to collect information on participants’ sociodemographic characteristics (domicile place, household income, and education), lifestyle (smoking and physical activity), medical history (gestational diabetes and pregnancy-induced hypertension), parity, pre-pregnancy body weight, and eating behavior at baseline. Pre-pregnancy body mass index (BMI) was calculated as body weight (kg) divided by the square of measured standing height at baseline (m^2^) [27]. Family size was divided into two categories (<4 and ≥4 members). Sufficient leisure-time physical activity was defined as at least 150 min/week of moderate activity or 75 min/week of vigorous activity or an equivalent combination [28]. Passive smoking was defined as exposure to others’ smoking for at least one day/week. Eating at home was specifically defined as consuming meals at home every day.

### 2.5. Statistical Analyses

Maternal characteristics were described as means (SDs) for continuous variables and *n* (%) for categorical variables. We imputed missing values using medians or modes if missingness is <5% of covariates, otherwise missing indicators were used.

Three multivariable logistic regression models were constructed to estimate the odds ratios (ORs) and 95% confidence intervals (CIs) of the association between salt intake and birth weight outcomes. Model 1 was adjusted for maternal age (year) and infant sex (male or female). Model 2 was additionally adjusted for maternal domicile location (northern China, Shanghai surrounding area, or southern China), pre-pregnancy BMI (<18.5, 18.5–23.9, or ≥24.0 kg/m^2^), height (cm), household income (<100,000, 100,000–350,000, or ≥350,000 yuan/year), education (<13 or ≥13 years), gestational periods at recruitment (first, second, or third trimester), passive smoking (never or ever), drinking (never or ever), and physical activity (<150 or ≥150 min/week) based on model 1. Model 3 further included eating at home (yes or no) and total energy intake (kcal/day) based on model 2. In the analysis of salt density as exposure, total energy intake was no longer considered as a covariate. For LBW and macrosomia, we further adjusted for gestational age at birth in the final model. *p* values for trends were obtained by modeling the median values of each category.

We also conducted prespecified subgroup analyses by maternal age, infant sex, baseline gestational periods, family size, passive smoking, physical activity, and eating at home to investigate the potential effect modification of each variable on the associations of salt intake with birth outcomes. Statistical significance of an interaction term between salt intake and the potential effect modifiers was used to evaluate interactions. To reduce the risk of false positive findings, we applied Bonferroni correction to the *p* values of interactions (*p* < 0.05 ÷ [7 subgroups × 4 outcomes]).

Finally, we performed the following sensitivity analyses to test the robustness of our findings. First, a series of exploratory models were conducted to examine potential confounding from the following factors based on the final model: (1) a further adjusted model for fruit and vegetable intakes, which are important sources of potassium that may counteract the effects of a high-salt diet; (2) additionally adjusted models for multivitamin, calcium tablets, and folic acid supplements; (3) an additionally adjusted model for gestational diabetes, which could lead to changes in dietary habits; (4) an additionally adjusted model for sodium from foods to examine the independent effects of sodium added during cooking; and (5) an additionally adjusted model for parity, as it may influence lifestyle choices during pregnancy [29]. Second, we excluded participants with pregnancy-induced hypertension at baseline. Third, we limited analyses to natural labor to examine the influence of delivery mode. Fourth, we limited analyses to term (born between 37 to <42 completed weeks) infants. Fifth, we filled missing data using multiple imputations to evaluate the impact of imputation methods on findings by creating 5 imputed datasets and pooling the estimates from logistic regression models across the imputed datasets [30]. Finally, we analyzed salt intake as a tertile variable.

A two-sided *p* value < 0.05 was considered statistically significant unless otherwise noted. All statistical analyses were performed in R (v4.2.1, R Foundation for Statistical Computing, Vienna, Austria).

### 2.6. Role of the Funding Source

The funders of the study had no role in study design, data collection, data analysis, data interpretation, or the writing of the report.

## 3. Results

### 3.1. Basic Characteristics of Participants

Basic characteristics of the participants by salt intake were presented in Table 1 and by tertiles of salt density in Appendix A. The study participants had a mean (SD) age of 29.6 (4.4) years and a mean BMI of 21.4 (2.9) kg/m^2^. The mean (SD) of salt intake was 5.9 (4.7) g/day, and the mean (SD) of estimated salt density was 3.4 (2.9) mg/kcal. The proportions of participants with a salt intake <5.0 g/day were 52.2%, 34.3% for 5.0–10.0 g/day, and 13.5% for ≥10.0 g/day. Participants with a higher salt intake were more likely to be younger and recruited during the second trimester, come from the northern regions, have lower education and income, eat at home, and have higher intakes of energy and vegetables (Table 1).

### 3.2. Salt Intake and Birth Weight Traits

We documented 106 LBW (2.5%), 234 macrosomia (5.5%), 250 SGA (5.9%), and 735 LGA (17.2%) cases. Associations between salt intake and the risks of birth weight outcomes were shown in Table 2. In the maternal-age-and-infant-sex-adjusted model (model 1), a higher salt intake was significantly associated with higher risks of LBW and SGA. The associations were stronger after further adjustment for other socioeconomic and lifestyle factors (model 2) or eating at home and total energy intake (model 3). In the final model, multivariable-adjusted ORs of LBW were 1.72 (95% CI 1.01–2.91) for participants consuming 5.0–10.0 g/day of salt intake and 2.06 (95% CI 1.02–4.13) for ≥10.0 g/day compared to those with <5.0 g/day (*p*-trend = 0.04). For SGA, ORs were 1.46 (95% CI 1.09–1.97) for 5.0–10.0 g/day and 1.69 (95% CI 1.16–2.47; *p*-trend = 0.006) for ≥10.0 g/day. No significant associations were observed for salt intake in relation to macrosomia or LGA.

Higher salt density was also associated with higher risks of LBW and SGA (Table 2), while no associations were observed for macrosomia or LGA. Specifically, ORs comparing extreme tertile of salt density were 1.91 (95% CI 1.08–3.36; *p*-trend = 0.01) for LBW and 1.63 (95% CI 1.18–2.25; *p*-trend < 0.001) for SGA in the final model.

### 3.3. Subgroup Analyses

The associations of salt intake and salt density with LBW and SGA were consistently observed across subgroups stratified by maternal age, infant sex, baseline gestational periods, family size, passive smoking, physical activity, and eating at home (Figure 1 and Figure 2; Appendix A).

### 3.4. Sensitivity Analyses

The findings were largely found in sensitivity analyses that assessed the effect of other potential confounding factors, missing covariates, and exposure reclassification (Appendix A). Among spontaneous labors (N = 2251), the association between salt intake and LBW was not significant (OR comparing ≥10.0 g/day with <5.0 g/day, 2.14; 95% CI 0.72–6.37; *p*-trend = 0.15, Appendix A).

## 4. Discussion

In this prospective cohort study of pregnant women in China, cooking salt intake above 5 g/day was associated with significantly higher risks of LBW and SGA. These findings were independent of potential confounding factors and remained stable in sensitivity analyses and stratified analyses.

To our knowledge, this is the first large prospective cohort study to examine the associations between salt intake and LBW, using a weighted method of household condiment consumption over a seven-day period. Two studies with sample sizes of less than 1000 participants, one conducted by Lee et al. in Australia and another by Birukov et al. in Denmark, observed no significant association of maternal sodium exposure with birth weight [4,14]. In addition, a recent study by Bank et al., which involved 7458 participants in the United States, found no significant differences in birth weight or SGA risk based on salt intake group [15]. Lee et al. and Bank et al. evaluated sodium intake with self-reported data from an FFQ, which is susceptible to known recall bias [31]. Correlation coefficients of sodium intake estimated from FFQ and urinary measures tend to be <0.37 [32]. In contrast, Birukov et al. used a single 24 h urine sample, which is inadequate for capturing the participant’s typical sodium intake due to substantial day-to-day variations [8]. Balance studies under controlled conditions suggest that the likelihood of the estimated salt intake from a single 24 h urine sample being within 3 g of the actual salt intake is 49% [33]. Moreover, the study conducted by Birukov et al. was subject to selection bias, as women who provided 24 h urine samples had higher pre-specified risk factors for gestational diabetes mellitus, such as overweight and glucosuria detected during pregnancy. Notably, salt sensitivity is higher in Asians than in Europeans, which may lead to a stronger association between salt intake and health consequences [34]. Therefore, findings from our study warrant confirmation in other populations.

Our findings are supported by several lines of evidence that collectively supported an inverse association between sodium exposure and birth weight. In a small clinical trial (*n* = 142) among pregnant women in South Korea, 16-week nutritional education on salt reduction led to 1140 mg/day lower maternal sodium intake, 277 g higher birth weight, and a lower proportion of infants with birth weights below 3000 g compared to the control group [35]. A pooled analysis of 24,861 mother–child pairs from 7 European cohorts also showed the Dietary Approaches to Stop Hypertension score, characterized by lower sodium intake, was associated with higher birth weight and lower risk of delivering LBW and SGA infants [16]. In addition, dietary patterns rich in high sodium foods, such as pizza and processed meat, have been associated with lower birth weight and a higher risk of SGA [17,18]. In light of these findings, further studies should examine the association of sodium and salt with birth weight, as well as optimal doses of salt intake during pregnancy.

Potential mechanisms that account for the observed associations are unclear. Previous studies showed that excessive salt intake was positively associated with risks of pregnancy-induced hypertension and pre-eclampsia, which are established risk factors of LBW or SGA [6,36,37]. In our study, the results were substantially unchanged when we excluded participants with pregnancy-induced hypertension at baseline, suggesting that other independent mechanisms may exist. It has been found that a high salt intake resulted in the inactivation of the renin–angiotensin–aldosterone system [38] which is essential to placental growth and fetal development. Both animal and human studies observed a positive relationship between plasma aldosterone levels and placental sizes. Pregnant mice with lower levels of aldosterone exhibited reduced fetal umbilical blood flow and smaller fetuses [39,40]. Additionally, a high salt intake has been associated with an increased risk of diabetes in the general population [41], while gestational diabetes is a known risk factor of macrosomia [42]. This indicates that gestational diabetes may modulate these relationships. However, we did not find an association between high salt intake and a higher risk of macrosomia or LGA (Table 2), even after adjusting for baseline gestational diabetes (Appendix A). These data underscored the complexity of these interactions, and clearly showed that our observational data needs confirmation by more experimental evidence. Collectively, current evidence highlights the need for further clinical or animal studies to explore the biological explanations for our findings.

Several limitations of our study must be noted. First, salt intake could be underestimated using cooking salt and soy sauce, and certainly does not fully reflect sodium exposure. It has been found that other foods, such as pickles and salted eggs, contributed to 5% of salt intake (3.6% total sodium) among Chinese people, and 7% of the sodium came from other sources including monosodium glutamate, soda, noodles, and stuffed buns [21]. Second, personal salt intake was estimated by averaging family condiment consumption and without accounting for the amount of salt from eating-out occasions that has been rapidly growing [43], which might be non-differential and attenuate the expected findings. In addition, findings were consistent when we restricted the analysis to participants who had all meals at home or when the analysis was stratified by family size. Third, the lack of data on low-salt soy sauce in this cohort may lead to an overestimation of sodium intake from soy sauce. However, given that only 6.4% of sodium intake is attributed to soy sauce and the limited salt reduction practices in China, such overestimation is likely to have minimal impact on our results. Fourth, salt intake in our study was estimated over one week, which was not sufficiently representative of usual salt intake owing to the potential day-to-day variations in salt intake. Previous validation studies among the Chinese population reported a strong correlation (r = 0.58) of dietary sodium estimated by combining household condiment-weighing and 24 h dietary recalls, with sodium excretions measured in 24 h urine samples over three consecutive days [44]. Fifth, we cannot exclude the possibility of residual confounding from other sources, such as gestational weight gain. Sixth, our study was based on pregnant women in Shanghai, which is a developed area in China; hence, the generalizability of results to other areas or ethnic populations is unknown. Finally, as in any observational studies, causal inferences cannot be made in this study.

## 5. Conclusions

In summary, we found that a daily cooking salt intake above the recommended amount during pregnancy is associated with increased risks of LBW and SGA in a large cohort of pregnant women in China. Whether salt reduction among women with an excessive intake may improve maternal and infant health warrants systematical evaluations through high-quality observational studies and clinical trials.

## Figures and Tables

**Figure 1 nutrients-17-00642-f001:**
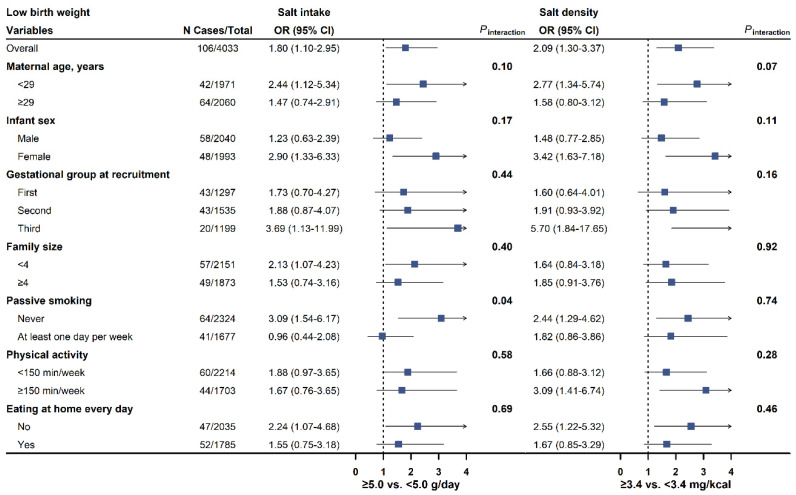
Forest plots for the association of salt intake (left, <5.0 g/day as reference) and salt density (right, <3.4 mg/kcal as reference) with the risk of low birth weight in pre-specified subgroups. ORs and 95% CI were calculated in logistic model adjusted for maternal age (continuous variables, in years), infant sex (male or female), maternal domicile (northern China, surrounding area of Shanghai, or southern China), baseline gestational periods (first, second, or third trimester), maternal education (<13 or ≥13 years), annual family income (<100,000, 100,000–350,000, or ≥350,000 yuan/year), pre-pregnancy BMI (<18.5, 18.5–23.9, or ≥24.0 kg/m^2^), height (continuous variables, in cm), passive smoking during pregnancy (never or ever), drinking (never or ever), physical activity (<150 or ≥150 min/week), eating at home every day (yes or no), energy intake per day (continuous variables, in kcal/day, not for salt density) and gestational week at birth (continuous variables, in weeks). N Cases/Total = number of cases and total participants in the study. OR = odds ratio. CI = confidence interval.

**Figure 2 nutrients-17-00642-f002:**
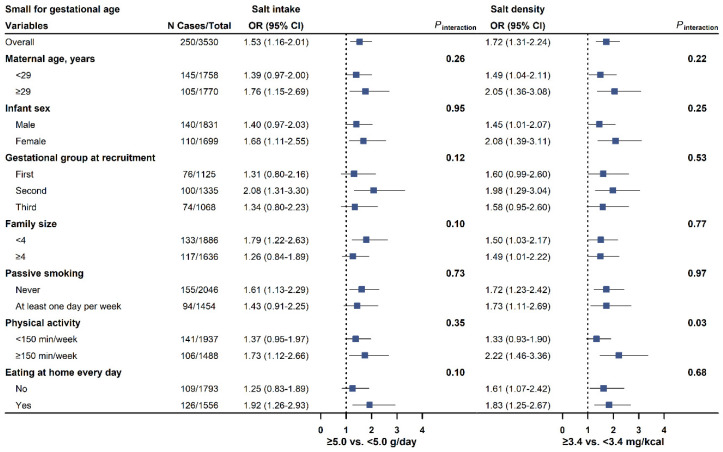
Forest plots for the association of salt intake (left, <5.0 g/day as reference) and salt density (right, <3.4 mg/kcal as reference) with the risk of small for gestational age in pre-specified subgroups. ORs and 95% CI were calculated in the logistic model adjusted for maternal age (continuous variables, in years), infant sex (male or female), maternal domicile (northern China, surrounding area of Shanghai, or southern China), baseline gestational periods (first, second, or third trimester), maternal education (<13 or ≥13 years), annual family income (<100,000, 100,000–350,000, or ≥350,000 yuan/year), pre-pregnancy BMI (<18.5, 18.5–23.9, or ≥24.0 kg/m^2^), height (continuous variables, in cm), passive smoking during pregnancy (never or ever), drinking (never or ever), physical activity (<150 or ≥150 min/week), eating at home every day (yes or no), and energy intake per day (continuous variables, in kcal/day, not for salt density). N Cases/Total = number of cases and total participants in the study. OR = odds ratio. CI = confidence interval.

**Table 1 nutrients-17-00642-t001:** Characteristics of pregnant women by salt intake in Shanghai, China.

	Salt Intake				
	Overall	<5.0 g/day	5.0–10.0 g/day	≥10.0 g/day	*p*
N	4267	2228	1465	574	
Maternal age, mean (SD), years	29.6 (4.4)	29.8 (4.3)	29.5 (4.5)	28.8 (4.6)	<0.001
Infants sex, male, *n* (%)	2190 (51.3)	1160 (52.1)	753 (51.4)	277 (48.3)	0.27
Maternal domicile location, *n* (%)					0.03
Northern China	695 (16.3)	360 (16.2)	234 (16.0)	101 (17.6)	
Shanghai surrounding area	2789 (65.4)	1496 (67.1)	928 (63.3)	365 (63.6)	
Southern China	783 (18.4)	372 (16.7)	303 (20.7)	108 (18.8)	
Baseline gestational periods, *n* (%)					<0.001
First trimester	1369 (32.1)	801 (36.0)	400 (27.3)	168 (29.3)	
Second trimester	1630 (38.2)	804 (36.1)	586 (40.0)	240 (41.8)	
Third trimester	1268 (29.7)	623 (28.0)	479 (32.7)	166 (28.9)	
Maternal Education, <13 years, *n* (%)	1316 (30.8)	569 (25.5)	525 (35.8)	222 (38.7)	<0.001
Household income, yuan/year, *n* (%)					<0.001
<100,000	727 (17.0)	310 (13.9)	295 (20.1)	122 (21.3)	
100,000–350,000	3041 (71.3)	1603 (71.9)	1029 (70.2)	409 (71.3)	
≥350,000	499 (11.7)	315 (14.1)	141 (9.6)	43 (7.5)	
Pre-pregnancy BMI, kg/m^2^, *n* (%)					0.48
<18.5	553 (13.0)	299 (13.4)	174 (11.9)	80 (13.9)	
18.5–23.9	3002 (70.4)	1551 (69.6)	1055 (72.0)	396 (69.0)	
≥24.0	712 (16.7)	378 (17.0)	236 (16.1)	98 (17.1)	
Maternal height, mean (SD), cm	160.9 (5.2)	161.0 (5.1)	160.7 (5.2)	160.8 (5.3)	0.24
Alcohol drinking, never, *n* (%)	3842 (90.0)	2002 (89.9)	1314 (89.7)	526 (91.6)	0.38
Passive smoking, never, *n* (%)	2488 (58.3)	1311 (58.8)	843 (57.5)	334 (58.2)	0.73
Leisure-time physical activity, *n* (%)					0.09
<150 min/week	2347 (55.0)	1213 (54.4)	794 (54.2)	340 (59.2)	
≥150 min/week	1920 (45.0)	1015 (45.6)	671 (45.8)	234 (40.8)	
Gestational age at birth, weeks	38.9 (1.3)	38.9 (1.3)	38.9 (1.4)	39.0 (1.3)	0.07
Eating at home every day, yes, *n* (%)	1898 (44.5)	738 (33.1)	818 (55.8)	342 (59.6)	<0.001
Total energy intake, mean (SD), kcal/day	1890.6 (725.2)	1791.1 (694.2)	1958.1 (722.4)	2104.4 (783.2)	<0.001
Fruits, mean (SD), g ^a^	281.4 (226.0)	275.5 (226.3)	283.6 (216.9)	298.6 (246.0)	0.08
Vegetables, mean (SD), g ^a^	221.1 (183.8)	213.8 (178.6)	228.2 (189.8)	230.9 (187.1)	0.03
Sodium, mean (SD), mg ^a^	2804.5 (1432.0)	2798.9 (1411.6)	2826.6 (1452.8)	2770.0 (1458.5)	0.70
Multivitamin, yes, *n* (%)	1782 (41.8)	935 (42.0)	603 (41.2)	244 (42.5)	0.82
Calcium, yes, *n* (%)	1752 (41.1)	870 (39.0)	643 (43.9)	239 (41.6)	0.01
Folic acid, yes, *n* (%)	2636 (61.8)	1399 (62.8)	873 (59.6)	364 (63.4)	0.10
Gestational diabetes, yes, *n* (%)	244 (5.7)	114 (5.1)	92 (6.3)	38 (6.6)	0.20
Pregnancy-induced hypertension, yes, *n* (%)	106 (2.5)	52 (2.3)	40 (2.7)	14 (2.4)	0.75
Nulliparity, yes, *n* (%)	2482 (58.2)	1307 (58.7)	821 (56.0)	354 (61.7)	0.05
Natural labor, yes, *n* (%)	2251 (52.8)	1190 (53.4)	746 (50.9)	315 (54.9)	0.18
Gestational age at delivery, *n* (%)					0.004
Pre-term	170 (4.0)	97 (4.4)	58 (4.0)	15 (2.6)	
Term	4077 (95.5)	2124 (95.3)	1402 (95.7)	551 (96.0)	
Post-term	20 (0.5)	7 (0.3)	5 (0.3)	8 (1.4)	

Abbreviations: SD, standard deviation. ^a^ The intake of fruits, vegetables, and sodium was assessed through the Food Frequency Questionnaire.

**Table 2 nutrients-17-00642-t002:** Associations of salt intake and salt density with birth outcomes among pregnant women in Shanghai, China ^a^.

	Salt Intake (g/day)				Salt Density (mg/kcal) ^b^			
Outcome	<5.0	5.0–10.0	≥10.0	*p*-Trend ^c^	T1 (<1.9)	T2 (1.9 to 3.7)	T3 (≥3.7)	*p*-Trend
** Low birth weight** ^g^								
N cases/Total	47/2122	42/1372	17/539		31/1347	32/1342	43/1344	
Model 1 ^d^	1.00 (ref)	1.41 (0.92–2.15)	1.49 (0.85–2.62)	0.13	1.00 (ref)	1.04 (0.63–1.72)	1.45 (0.90–2.31)	0.09
Model 2 ^e^	1.00 (ref)	1.75 (1.04–2.94)	2.13 (1.08–4.21)	0.02	1.00 (ref)	0.96 (0.52–1.77)	1.99 (1.14–3.48)	0.006
Model 3 ^f^	1.00 (ref)	1.72 (1.01–2.91)	2.06 (1.02–4.13)	0.04	1.00 (ref)	0.93 (0.50–1.73)	1.91 (1.08–3.36)	0.01
** Small for gestational age**								
N cases/Total	105/1861	99/1191	46/478		68/1174	67/1176	115/1180	
Model 1	1.00 (ref)	1.49 (1.12–1.98)	1.71 (1.19–2.46)	0.003	1.00 (ref)	0.98 (0.69–1.38)	1.69 (1.23–2.31)	<0.001
Model 2	1.00 (ref)	1.50 (1.12–2.01)	1.73 (1.20–2.51)	0.003	1.00 (ref)	0.99 (0.69–1.40)	1.70 (1.24–2.34)	<0.001
Model 3	1.00 (ref)	1.46 (1.09–1.97)	1.69 (1.16–2.47)	0.006	1.00 (ref)	0.96 (0.68–1.37)	1.63 (1.18–2.25)	<0.001
** Macrosomia** ^g^								
N cases/Total	106/2181	93/1423	35/557		75/1391	80/1390	79/1380	
Model 1	1.00 (ref)	1.37 (1.03–1.83)	1.35 (0.91–2.00)	0.09	1.00 (ref)	1.08 (0.78–1.50)	1.08 (0.78–1.50)	0.68
Model 2	1.00 (ref)	1.34 (1.00–1.80)	1.28 (0.85–1.93)	0.17	1.00 (ref)	1.08 (0.77–1.50)	1.02 (0.73–1.42)	>0.99
Model 3	1.00 (ref)	1.29 (0.96–1.75)	1.22 (0.80–1.85)	0.30	1.00 (ref)	1.05 (0.76–1.47)	0.98 (0.69–1.38)	0.83
** Large for gestational age**								
N cases/Total	367/2123	274/1366	94/526		248/1354	246/1355	241/1306	
Model 1	1.00 (ref)	1.21 (1.02–1.44)	1.07 (0.83–1.37)	0.40	1.00 (ref)	1.00 (0.82–1.21)	1.04 (0.85–1.26)	0.69
Model 2	1.00 (ref)	1.20 (1.00–1.43)	1.04 (0.81–1.34)	0.55	1.00 (ref)	1.00 (0.82–1.22)	1.03 (0.84–1.25)	0.79
Model 3	1.00 (ref)	1.15 (0.96–1.38)	0.98 (0.75–1.27)	0.97	1.00 (ref)	0.98 (0.80–1.20)	0.99 (0.81–1.22)	0.98

Abbreviations: N Cases/Total = number of cases and total participants in the study. OR = odds ratio. I = confidence interval. Ref = reference. *p*-trend = *p* value for trend. ^a^ ORs and 95% CIs for the association were calculated in logistic models. ^b^ The first and second tertiles of salt density in the population are 1.9 and 3.7 mg/kcal, respectively. ^c^ The *p* values for trends were obtained through modeling the median value of each category into the logistic regression models. ^d^ Model 1, adjusted for maternal age (continuous variables, in years) and infant sex (male or female). ^e^ Model 2, additionally adjusted for maternal domicile (northern China, surrounding area of Shanghai, or southern China), baseline gestational periods (first, second, or third), maternal education (<13 or ≥13 years), annual family income (<100,000, 100,000–350,000, or ≥350,000 yuan/year), pre-pregnancy BMI (<18.5, 18.5–23.9, or ≥24.0 kg/m^2^), height (continuous variables, in cm), passive smoking during pregnancy (never or ever), drinking (never or ever), physical activity (<150 or ≥150 min/week) based on model 1. ^f^ Model 3, further adjusted for eating at home every day (yes or no) and energy intake per day (continuous variables, kcal/day) based on model 2. ^g^ Analyses for low birth weight and macrosomia were additionally adjusted for gestational week at birth in the final model.

## Data Availability

The data during the current study are available from the corresponding author on reasonable request.

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
