# Peer review of "Associations of Cooking Salt Intake During Pregnancy with Low Birth Weight and Small for Gestational Age Newborns: A Large Cohort Study"

_nutrients, 2025, doi:10.3390/nu17040642_

Round 1

Reviewer 1 Report

Comments and Suggestions for Authors

This is an interesting and important topic utilising a cohort already available. I think it would be important to note if anyone declined to take part in the study. Does anyone in China use salt reduced soy sauce and were the women included all nulliparous?

I noted a few English errors and ambiguities: line 71 instead of among this population among the Chinese population as western countries were mentioned immediately preceding. Line 249 leave out "participants with", line 276 led not lead, line 302 buns not bons

Author Response

Thanks for your thorough review of our manuscript. We have provided point-by-point responses to your comments, and please see the attachment.

Reviewer 2 Report

Comments and Suggestions for Authors

I have read this paper with great interest, and with a background on clinical perinatal research. I assess the data as reported as valuable to the research community. I hereby highly value the consistent use of association between salt intake and birth weight, as I still struggle to understand the potential underlying mechanisms. The estimations, not similar to observed values, and the overall ‘low adjusted OR’ further add to the cautious approach needed. In contrast, the absence of an association with LGA or macrosomia is not unexpected, as this is very likely mainly modulated by gestational diabetes.

It is obviously well known that maternal hypertension is associated with birth weight outcome variables, while it is commonly claimed that this hypertension is mainly ‘increased resistance’ modulated, less with volume overload. Do you have data on pregnancy-related hypertension or preeclampsia, or well known modulators, like smoking. In the discussion (line 286 onwards) you suggest exclusion of pregnancy-induced hypertension, but this could it itself be a bias as this will more commonly appear in the third or second trimester, not in the first trimester.

Do you have information on the incidence of gestational diabetes in this cohort, or in this region?

If I understood the recruitment strategy well, there was ‘only’ one point measurement in each pregnant volunteer, and no longitudinal data in the same case ? As you have collected data in the first, second and third trimester, is there any effect of the trimester on the outcomes of interest ?

Line 87: perhaps ‘if hey subsequently had not delivered a live, singleton baby to make it clear that this info only because afterwards available ?

Was salt intake also associated with gestational age at delivery, or type of delivery  (table 1 suggest that this is not the case, but please confirm)

As this is relevant to the outcomes, how has the duration of gestational age been documented (early ultrasound, last menstrual period, or clinical ?)

3.2

Your ‘incidence’ of LBW or SGA is lower that expected (10%). Could this reflect bias, or how do the authors assess these results in light of the research question involved.

Line 253: ‘strong’ is in my opinion overinterpretation.

In the IRB statement, I miss the individual volunteers informed consent aspect. Please add.

Author Response

Thank you for your valuable feedback on our manuscript. Please see the attachment.

Reviewer 3 Report

Comments and Suggestions for Authors

This study about the association of maternal salt intake with Low Birth Weight and Small for Gestational Age children is interesting because it highlights the connection between maternal diet during pregnancy and infant weight.

The article is well written, complete in its different sections and would benefit only from few corrections:

1.     The title could be modified as follows: “Association of Maternal Salt Intake with Low Birth Weight and Small for Gestational Age Newborns: a Large Cohort study”.

2.     The introduction section is adequately written, providing essential information to the readers.

3.     The material and methods section is well written, ordered in key sections.

4.     The results section is well articulated. Anyway, Authors should add numbers and p-values related to the sentence “Participants with higher salt intake were more likely to come from the 176 northern regions, have lower education and income, eating at home, and have higher in-177 takes of fruit and vegetables”. Maybe it could be useful to add a Table as supplementary material, for example.

5.     The discussion section is well written and argued.

6.     Tables and graphics are clear and prompt.

Author Response

Thank you very much for affirming our research. We have carefully addressed each of your comments in detail. Please see the attachment.

Round 2

Reviewer 2 Report

Comments and Suggestions for Authors

the comments and concerns have been addressed, nothing to add

Author Response

Thank you for your review and feedback!